# Impact of Antiretroviral Therapy on Oral Health among Children Living with HIV: A Systematic Review and Meta-Analysis

**DOI:** 10.3390/ijerph191911943

**Published:** 2022-09-21

**Authors:** Phoebe Pui Ying Lam, Ni Zhou, Cynthia Kar Yung Yiu, Hai Ming Wong

**Affiliations:** 1Paediatric Dentistry, Faculty of Dentistry, The University of Hong Kong, Pokfulam, Hong Kong; 2School of Stomatology, Kunming Medical University, Kunming 650032, China

**Keywords:** HIV, adolescents, antiretroviral therapy, oral health, children, meta-analysis

## Abstract

Oral health is an integral component of general health and well-being but might be undermined among children living with HIV (CLWH) due to the condition itself or the antiretroviral therapy (ART) received. This review summarises the current evidence and compares the oral health status of the CLWH who were treatment-naïve with those undergoing different ART medications. Fourteen studies were included in the final qualitative and quantitative analyses. This review identified no significant difference in the prevalence of caries, periodontal conditions, and tooth development between both groups. Orofacial opportunistic infections were more prevalent in the CLWH without ART. Children undergoing ART with a duration longer than 3 years had a significantly lower prevalence of oral candidiasis and CD4+ T-cell counts. However, due to the insufficient number of well-administered case–control studies with adequate sample size, the quality of the evidence in all outcomes was of very low certainty.

## 1. Introduction

Oral health contributes to our general well-being and overall health. As an integral part of our body, our mouth and teeth enable us to enjoy food, express ourselves, socialise with others, and establish our own personal identity and self-esteem [1]. The consequences of oral diseases would profoundly impact the quality of life of both adults and children [2]. Untreated dental caries and other oral pathologies can result in pain, disturbed eating and sleeping, and inability to perform schoolwork and daily tasks [2]. Prolonged repercussions, for instance, reduced growth and diminished weight gain, would significantly undermine children with lifelong implications [3]. Unfortunately, over 3.5 billion people globally are still burdened by preventable oral diseases and dental conditions [4].

HIV infection and AIDS have been another global health hazard since 1981. With increased HIV awareness campaigns and public health education targeting the populations at risk of HIV, the HIV incidence and vertical mother-to-child transmission have gradually declined [5,6]. In 2019, approximately 38 million people worldwide were infected with HIV, with 2.8 million being children and adolescents [7]. With the early onset of ART, most CLWH can live an asymptomatic life with minimal complications [8,9]. Nonetheless, the overall global ART coverage among children was only 28%, with around 1.9 million children still waiting for the appropriate treatment required [10]. Over 85% of children in West, Central, and North Africa were still treatment-naïve to life-saving ART [10].

Manifestations in the oral cavity are usually the early signs of HIV infections. The most common oral manifestations among the Asian adult population are oral candidiasis (37.7%), followed by mucosal hyperpigmentation (22.8%), oral hairy leukoplakia (10.1%), oral malignancies (2–4%), and necrotising ulcerative periodontal diseases (4.2–7.6%) [11].

ART may also result in undesirable side effects, which are detrimental to oral health. Orofacial manifestations and the opportunistic infections of AIDS might still be found among individuals living with HIV treated with ART. At the same time, the susceptibility of patients with HIV to dental caries, periodontal conditions, and other oral diseases might be elevated due to the infection itself or the drawbacks of ART. The major classes of ART agents, including protease inhibitors, didanosine, and lamivudine–zidovudine, are known to induce xerostomia [12]. Medications given to young children in a form of sweetened liquid might escalate their risk of dental caries [13]. The presence of oral ulcerations might also make oral hygiene maintenance difficult.

According to the United Nations Convention on the rights of the child, all children deserve the right to the best possible health [14]. By understanding the oral health status of CLWH with or without ART, more cost-effective health care policies can be implemented to improve their oral and general health. Our review aimed to summarise all the currently available evidence to compare the oral health status of CLWH undergoing different ART treatments with treatment-naïve controls.

## 2. Materials and Methods

### 2.1. PECO(S) Statements

The systematic review and meta-analyses performed in this study were administered and reported following the PRISMA guidelines [15]. The review was registered on PROSPERO (registration number CRD42019148245) while following the respective PECO(S) questions.

### 2.2. Participants (P)

Participants were children and adolescents under the age of 18 years old, as defined by the United Nations, who were infected by HIV.

### 2.3. Exposure (E) and Control (C)

The exposure group was the CLWH undergoing ART treatment, while the CLWH who were treatment-naïve were considered the control group. When comparisons were made among the CLWH receiving the different types of ART, those undergoing combined antiretroviral therapy (cART) were considered the exposure group, whereas those receiving conventional HIV medications, e.g., monotherapy and dual-therapy, served as the control.

### 2.4. Outcome Measures (O)

Outcome measures included the following factors: (a)Dental caries;(b)Oral hygiene and periodontal health status;(c)HIV-related orofacial diseases based on the WHO clinical staging and immunological classification, including stage 2 (angular cheilitis, herpes zoster, linear gingival erythema, recurrent oral ulceration, parotid enlargement), stage 3 (oral candidiasis, oral hairy leukoplakia acute necrotising ulcerative gingivitis/periodontitis), and stage 4 (herpes simplex infection, Kaposi’s sarcoma);(d)Saliva immunoglobulin quantity;(e)Oral microbiome count;(f)Dental development and maturation.

### 2.5. Studies (S)

Case–control observational studies and randomised controlled studies with full-text reports available in English were included. Cross-sectional studies or interventional studies without control groups were excluded.

### 2.6. Search Strategies

Four electronic databases (Ovid Embase, Ovid MEDLINE, PubMed, and Scopus) were systematically searched from inception to 3 August 2022. A strategic scrutiny of Medical Subject Heading (MeSH) terms and broad keywords was implemented, which is presented in the Appendix A. The reference lists of the previous relevant literature reviews were also screened through manual searches, to ensure no potentially eligible studies were omitted.

### 2.7. Selection of Studies

A dual, independent screening method was performed, with two reviewers (P.P.Y.L and N.Z.) first reviewing the titles and abstracts of all the retrieved reports. The full texts of any relevant studies identified were further retrieved and assessed for final eligibility. Any disagreement was resolved by discussion or consultation with a third reviewer (H.M.W.). The inter-examiner reliability was determined with Cohen’s kappa coefficient (κ).

### 2.8. Data Extraction and Management

Data aggregation and documentation from the included reports were independently performed by the two reviewers, utilising a standardised data extraction form. The extracted data included information on the participants (location of recruitment, gender, and age distribution), the exposure and control groups (diagnostic methods, inclusion and exclusion criteria, confounders matched), and the outcomes (caries prevalence and severity, oral hygiene and periodontal status, HIV-related orofacial diseases, saliva immunoglobulins quantity, dental development, and other oral manifestations).

### 2.9. Measures of Effect

The prevalence of dental caries and HIV-related orofacial diseases were categorised into dichotomous outcomes and evaluated based on the odds ratios (ORs) and confidence intervals (CIs) obtained. The adjusted OR was used when there was no event in at least one group, in which a fixed value (0.5) was added to all the cells in the 2 × 2 tables to overcome any computational error [16].

For the continuous outcomes, the means and standard deviations were obtained and compared. These included (1) caries, measured by the number of decayed, missing due to decay, or filled teeth and the surfaces in permanent dentition (DMFT/DMFS) and primary dentition (dmfs/dmfs); (2) the plaque index (PI) and the gingival index (GI) [17,18]; and (3) the salivary flow rate, pH, and antibody levels.

### 2.10. Subgroup Analyses

To control for potential confounders, subgroup analyses were performed to assess their potential influence on the effect estimates. The confounders evaluated included the stages of dentitions (permanent, mixed, and primary dentition), CD4+ T-cell counts, and other immunological factors, durations, and types of medication, if available.

### 2.11. Assessment of Risk of Bias 

The validity of each included study was assessed with the Risk Of Bias in Non-Randomised Studies of Interventions tool (ROBINS-I tool) [19]. The ROBINS-I tool consists of 7 domains, namely (I) bias due to confounding, (II) bias in the selection of participants for the study, (III) bias in the classification of interventions, (IV) bias due to deviation from the intended interventions, (V) bias due to the missing data, (VI) bias in the measurement of outcomes, and (VII) bias in the selection reported. Guided by the signalling questions in each domain, the rating of the risk of bias in each domain and at an overall level were classified and assessed as low, moderately serious, or critical.

### 2.12. Data Synthesis

The meta-analyses were performed using Stata version 13.1 (StataCorp, College Station, TX, USA, 2013). Those meta-analyses including fewer than five studies were evaluated with the fixed-effect model, whereas those with more than five studies were evaluated with the random-effect model [20]. Sensitivity analyses were performed if the studies with low validity were suspected to have significantly influenced the effect estimate. 

### 2.13. Assessment of Heterogeneity

Inconsistency in each meta-analysis was evaluated with I^2^ statistics and a chi-square test. Substantial heterogeneity was considered if I^2^ > 50% and *p* < 0.05 in the chi-square test [20].

### 2.14. Assessment of Publication Bias

When there were more than 10 studies that contributed to the meta-analyses, funnel plots were utilised to assess the presence of any potential publication bias in the current evidence [21]. However, funnel plots were not used in this systematic review and meta-analyses, as none of the outcomes included over 10 studies.

### 2.15. Assessment of Quality of Evidence

The Grading of Recommendations Assessment Development and Evaluation (GRADE) approach was used to assess the certainty of evidence [22]. All the outcomes included only observational studies and began with low certainty. If both reviewers raised any serious concerns regarding the risk of bias, imprecision, inconsistency, indirectness, and publication bias, the certainty of the evidence would further drop to very low.

## 3. Results

### 3.1. Study Selection

After duplicate removal, 783 records were identified from our strategic search of the literature. After the initial screening by titles and abstracts, the full reports of 86 studies were scrutinised. Fourteen studies were found eligible and contributed to either or both comparisons between the CLWH with and without ART, and the CLWH receiving different medications. The inter-examiner reliability determined with Cohen’s kappa coefficient (κ) was 0.948. The flow of the screening process and reasons for exclusion are reported in Figure 1.

### 3.2. Study Characteristics

Table 1 reports the characteristics of the 14 included studies [23,24,25,26,27,28,29,30,31,32,33,34,35,36]. A total of 3525 CLWH and adolescents, aged from three months to 17 years, were recruited from hospitals and AIDS outpatient centres in India, Brazil, and several Asian countries. Three studies [28,33,34] were cohort studies in which 2438 CLWH served as their own control individuals, with their pre-and post-ART oral health conditions being compared and reported. The remaining studies adapted a parallel group design, with 294 CLWH with no treatment being compared with 793 CLWH receiving different ART treatments.

### 3.3. Risk of Bias of Included Studies

All the studies were considered to have a low risk of bias in the following domains: (II) bias in the selection of participants for the study, (III) bias in the classification of interventions, (IV) bias due to deviation from the intended interventions, (V) bias due to the missing data and domain, and (VII) bias in the selection of the reported results [23,24,25,26,27,28,29,30,31,32,33,34,35,36].

However, all the studies were graded as having a moderate risk of bias in domain (VI) bias in the measurement of outcomes. This was because the outcome measures involved only visual examination, which could be influenced by the assessors’ judgement. However, no information was available to inform whether the outcome assessors were blinded to the medical history of the subjects. Nonetheless, as the assessment methods were comparable, and the misclassification of outcomes was minimally related to the medication status, both reviewers found all the studies sound for non-randomised studies in this domain but not comparable to a well-performed randomised trial.

In domain (I) bias due to confounding, 11 studies were evaluated as having a low risk of bias [23,24,25,26,27,28,29,30,31,33,36]. When evaluating the occurrence of orofacial opportunistic infections, nine studies considered confounders such as immunological status, CD4+ T-cell counts, and the duration of therapy [23,24,25,26,27,28,29,30,31,33]. Appropriate analysis methods were employed to evaluate the effects of confounders when assessing orofacial infections [23,24,25,26,27,28,29,30,31,33]. For one study evaluating dental caries, the confounders including age and sociodemographic factors were controlled by recruiting subjects in similar settings [36].

Three studies were considered to have a serious risk of bias, as important confounders were not accounted for in their comparisons. When evaluating the saliva IgA levels, two studies [32,34] did not report the caries status or any potential associated factors other than immunodeficient causes. When evaluating dental development, one study [35] did not adequately match the chronological age of each comparison pair of CLWH.

Based on the tool’s guidelines, three studies had one domain being graded as a serious risk of bias, thus being considered as studies of serious risk of the overall bias [32,34,35]. The reviewers determined the remaining nine studies to be of low risk of overall bias even though they were graded as having a moderate risk of bias in one domain regarding outcome assessment, as this was due to the insufficient information reporting the blinding of the outcome assessors (Figure 2).

### 3.4. CLWH under ART versus Those without Dental Caries Experience

Two studies evaluated and reported the outcome regarding dental caries but with different measurements [30,36]. When comparing the CLWH who were treatment-naïve and those undergoing ART, Ponnam et al. (2012) measured the caries prevalence of 190 CLWH [36], while Oliscovicz et al. (2015) assessed the number of decayed, missing, and filled teeth in the primary and permanent dentitions (dmft and DMFT) of 111 individuals [30]. Despite using different parameters, both studies identified no significant difference in terms of caries experience between the two groups (odds ratio, 0.91; 95% CI, 0.5–1.65; Appendix A) [30,36].

#### Oral Hygiene and Periodontal Status

Two studies assessed the prevalence of periodontal diseases among 250 CLWH. No significant difference in terms of the prevalence of periodontal diseases was found between the CLWH under medications and those without (odds ratio, 0.82; 95% CI, 0.46–1.44; I2 = 0.0%, *p* = 0.456) [30,36]. In comparison, no studies compared and reported the oral hygiene status of the two groups (odds ratio, 0.73; 95% CI, 0.13–4.26) (Appendix A and Figure 3).

### 3.5. Oral Candidiasis and Oral Microbiome Count

The subgroup analyses demonstrated that the prevalence of oral candidiasis was significantly lower in the CLWH undergoing ART (odds ratio, 0.35; 95% CI, 0.19–0.63) and cART (odds ratio, 0.35; 95% CI, 0.24–0.51) than that in treatment-naïve individuals, a result that also held true when comparing the group after cART treatment with the one prior to cART treatment (odds ratio, 0.19; 95% CI, 0.15–0.25) [23,24,27,28,32,33,36].

However, no significant difference was identified between treatment-naïve individuals and those under monotherapy or dual therapy. Considerable levels of heterogeneity were identified in most subgroup analyses and in the overall meta-analysis (I^2^ = 86.5%, *p* < 0.001; Appendix A and Figure 4) [23,24,27,28,32,33,36].

### 3.6. Other HIV-Related Orofacial Diseases

Other than oral candidiasis, parotid hypertrophy (odds ratio, 0.01; 95% CI, 0.00–0.05), ulcerative stomatitis (odds ratio, 0.03; 95% CI, 0.02–0.07), Kaposi’s sarcoma (odds ratio, 0.00; 95% CI, 0.00–0.03) and acute necrotising ulcerative gingivitis (odds ratio, 0.00; 95% CI, 0.00–0.02) were found to be significantly more prevalent among the CLWH who were treatment-naïve (Appendix A) [23,24,27,28,32,33,36].

Whereas no significant differences were found in terms of the prevalence of other orofacial opportunistic diseases between both groups, which included angular cheilitis (odds ratio, 0.58; 95% CI, 0.17, 1.99), linear gingival erythema (odds ratio, 1.29; 95% CI, 0.58–2.88), recurrent oral ulcerations (odds ratio, 1.50; 95% CI, 0.72, 3.13), oral hairy leukoplakia (odds ratio, 2.75; 95% CI, 0.39, 19.43), and chronic herpes simplex infection (odds ratio, 2.47; 95% CI, 0.88, 6.97) among the CLWH with and without ART or cART. No significant differences were identified in the prevalence of mucosal hyperpigmentation (odds ratio, 1.79; 95% CI, 0.89, 3.61) and mucocele (odds ratio, 0.02; 95% CI, 0.00–1.09) (Appendix A) [23,24,27,28,32,33,36].

### 3.7. Saliva Immunoglobulins Quantity

Divergent results were obtained from two studies that compared and reported the salivary IgA (SIgA) concentration of children prior to the onset of ART and those undergoing ART. However, neither of the two studies reported the caries status of the children in both groups.

Subramaniam (2015) reported that the SIgA levels in CLWH (6.2 ± 1.7 mg/dL) before ART treatment were significantly lower than the SIgA levels in those receiving ART (8.6 ± 2.9 mg/dL; *p* < 0.001) for more than 3 years, despite similar CD4 cell counts [34]. 

Pomarico (2009) identified significantly lower SIgA levels in those children undergoing ART than in those without ART (*p* = 0.05). However, the oral Candida carriage and salivary yeast concentration were also lower in the former group [32]. 

### 3.8. Dental Development and Maturation

Only one study provided the data for this comparison, employing Nolla’s method to assess dental development [35]. Trigueiro (2010) compared the CLWH taking four different combinations of medications with those without, with the number of participants ranging from 3 to 23 subjects. The authors reported that the mean differences between the dental age and the chronological age ranged from 5.35 to 14.36 months, (SE 3.03–15.86 months) for those taking nucleoside reverse transcriptase inhibitors, protease inhibitors, and non-nucleoside reverse transcriptase inhibitors, either alone or in combination, while that of those without medication was only −0.67 months [35].

### 3.9. CLWH under Different ART Medications

No eligible studies were found comparing the prevalence of angular cheilitis, oral hairy leukoplakia and dental development between the CLWH and adolescents taking different ART medications (Appendix A). 

No significant differences were found in terms of caries prevalence and severity [25,30], gingivitis [25,30], linear gingival erythema [30], recurrent oral ulceration [30], persistent parotid enlargement [30], acute necrotising ulcerative periodontitis [33], chronic herpes simplex infection [30], and Kaposi’s sarcoma [33] (Appendix A). 

Four studies compared the prevalence of oral candidiasis between the CLWH taking different medications [26,28,30,33]. Those taking cART had a significantly lower prevalence of oral candidiasis than those taking monotherapy or dual therapy (odds ratio, 0.22; 95% CI, 0.16, 0.29) (Appendix A and Figure 5) [26,28,30,33].

### 3.10. Other Associated Factors

CD4+ T-cell counts and the duration of the medications were two important confounders that might significantly affect the prevalence of orofacial opportunistic infections. 

For CD4+ T-cell counts, three out of four studies identified significantly more orofacial opportunistic infections among children with either less than 250 CD4+ T cells/mm^3^ or 15% CD4+ T cells [23,24,33]. The only study that did not identify CD4+ T-cell counts as a possible confounder utilised less than 200 rather than 250 CD4+ T cells/mm^3^ as the benchmark in their analyses [27].

As regards the duration of ART, two studies reported that the duration of ART did not pose a significant influence on the occurrence of orofacial opportunistic infections, except for candidiasis. However, the duration of ART was also positively associated with CD4+ T-cell counts, as those children taking ART for three years or more were less likely to have oral candidiasis and CD4+ T-cell counts less than 250 CD4+ T cells/mm^3^.

## 4. Discussion

The literature investigating the oral health conditions of CLWH was scarce. The number of well-administered trials providing head-to-head comparisons between the treatment-naïve CLWH and those undergoing different ART regimens was even more limited. Most outcomes were generated from very few studies with insufficient sample sizes. Most of the data gathered were heterogeneous and could not be aggregated for further analyses. Following the GRADE approach, due to serious concerns in imprecision and inconsistency, the certainty of the evidence informing the oral health status of CLWH was in general low or very low (Appendix A).

The CLWH were generally assumed to have poorer oral health and more caries experience. However, when compared with their unaffected peers, no significant differences between their permanent tooth caries experiences were identified [37]. When taking into account confounders such as socioeconomic status, it is also not surprising to find no significant differences between those treated with ART and those who were not. Poverty has been a common risk factor for both caries and HIV infections [38,39]. As lower education attainment is inversely associated with one’s knowledge to prevent the transmission and progression of both infections [38,39], socioeconomically disadvantaged groups were more disproportionately affected by both diseases.

Oral candidiasis was less prevalent among the CLWH receiving ART than those without, in particular cART. Oral candidiasis is one of the first few signs indicating immunosuppression or a failure of ART, which coincides with the findings that fewer children undergoing cART had less than 250 CD4+ T cells/mm^3^. The cART was found to better enhance CD4 cell counts and diminish the viral loads than monotherapy and dual therapy [40], as the children receiving the latter had a similar prevalence of oral candidiasis to those without therapy. These findings further affirmed the effectiveness of cART and why it is the current standard of care for those paediatric patients diagnosed with HIV infections [40].

However, inconsistent findings were found in the comparison results of SIgA concentrations between children with and without ART. SIgA is the main antigenic material found in saliva against oral pathogens [41]. The association between a reduction in CD4+ T-cell count and diminished SIgA is still debatable [42]. Additionally, other than candidiasis infections, the presence of caries, periodontitis, and other oral infections can also alter SIgA levels [43,44]. Due to the incomplete reporting of these confounders, valid conclusions cannot be drawn.

Inconsistent results were also identified when comparing the prevalence of mucosal hyperpigmentation among the CLWH undergoing ART and those without medications. One possible explanation is that both ART and HIV infection might be associated with increased melanin production [45,46]. The former was previously identified to elevate melanocyte-stimulating hormones, while the latter might induce immunopathologic changes in the oral mucosa. 

Adverse effects of ART on tooth eruption, bone development and remodelling have been reported in the literature [47,48,49]. The speculated theories include potential interference with vitamin D metabolism [50] and mitochondrial toxicity associated with NRTIs [51]. One of the included studies reported a delay in tooth mineralisation and dental development, but we found that the discrepancies between the chronological age and the dental age might also be attributed to the broad margins of errors in dental age assessments, especially due to variations in gender, ethnicities, and different age groups [52]. As the study included fewer than 10 subjects in each subgroup, with a broad age range from 37 to 168 months [35], the authors found the available evidence insufficient to draw a valid conclusion.

The oral manifestations of adults living with HIV (ALWH) receiving ART or being treatment-naïve are beyond the scope of this review, but increasing evidence has suggested the oral-health-related side effects of prolonged ART among adults. Despite inconsistent findings, an increased prevalence of periodontitis and alveolar bone loss in relation to osteoporosis and osteodystrophy have been reported among the ALWH receiving ART [53]. A strikingly higher prevalence of HPV-related oral lesions and neoplasms have also been observed among the ALWH, including squamous cell papillomas, condylomas, and focal epithelial hyperplasia [54,55,56]. However, these manifestations were not reported among children in the studies identified. A possible explanation might be that these conditions would only manifest at a later stage of life with a longer duration of ART. However, owing to the limited number of relevant studies identified, further research is required for validation and clarification.

Careful administration and reporting in accordance with the PRISMA guidelines and the Cochrane Handbook for Systematic Review of Interventions [15,21], the utilisation of ROBINS-I [19], qualitative and quantitative syntheses, and the GRADE assessment [22] are all strengths of our review. Rather than employing the usual assessment tool for assessing observational studies, for instance, the Newcastle–Ottawa Scale (NOS) [57], the ROBINS-I tool was used to evaluate the internal validity. Revised from the Cochrane RoB tool for randomised trials [21] and the QUADAS 2 tool [58], ROBINS-I provides a comprehensive algorithm with detailed guidance to determine the risk of biases in each aspect, allowing higher inter-reviewer agreements to be reached [59]. The results obtained from ROBINS-I can also be directly incorporated into the GRADE assessment, as the studies were judged on a similar scale to that of randomised controlled trials [60]. The requirements in ROBINS-I are also more demanding toward the information reported, reducing the risk of underscoring biased results, unlike NOS [59]. Another modification of ROBINS-I, ROBINS-E(Exposure) [61] was not used, as it was still under development and validation at the time of this study [61]. Since ROBINS-I was also found to be applicable to evaluate unintentional exposures by many researchers [61], the latter was chosen.

The limitations of this study include its inevitable exclusion of non-English translated reports; however, Jüni et al. (2002) and Moher et al. (2000) showed that such influence might not be as considerable as compared with language-inclusive meta-analyses [62,63]. This review also excluded those studies that included subjects over 18 or those in which the data could not be extracted. Meanwhile, meta-regression could not be conducted due to incomplete reporting and thus the inability to pull relevant data from the included studies. Due to the limited number of eligible studies included in this review, funnel plots could not be employed to evaluate publication bias.

## 5. Conclusions

When the CLWH who were undergoing ART were compared with those not taking medications, no significant difference in terms of caries prevalence and periodontal diseases was observed. Orofacial opportunistic infections, including oral candidiasis, acute necrotising ulcerative periodontal diseases, and Kaposi’s sarcoma, were found to be more prevalent among the CLWH without ART. The underlying factors for opportunistic infections might be attributed to the condition of less than 250 CD4+ T cells/mm^3^. No evidence has suggested that dental development and maturation status were influenced by HIV status or medications. As there is a still dearth of studies in the literature reporting the oral health status of the CLWH who are treatment-naïve or receiving different ART regimens, our review encourages high-quality research with a well-administered study design, analysis, and reporting to be performed. Hence, more concrete evidence regarding the oral health status of CLWH can be established.

## Figures and Tables

**Figure 1 ijerph-19-11943-f001:**
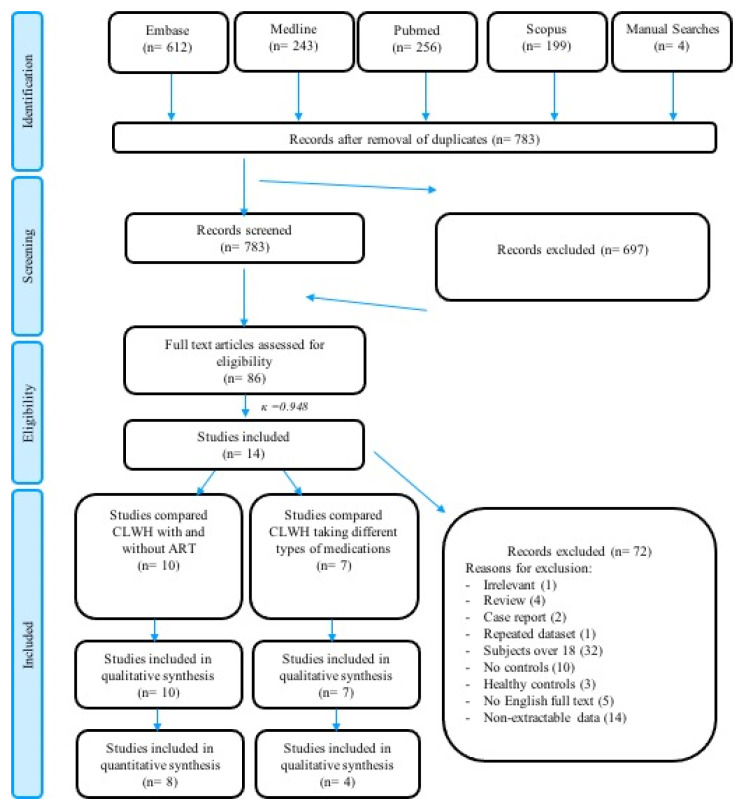
PRISMA flowchart of the current review.

**Figure 2 ijerph-19-11943-f002:**
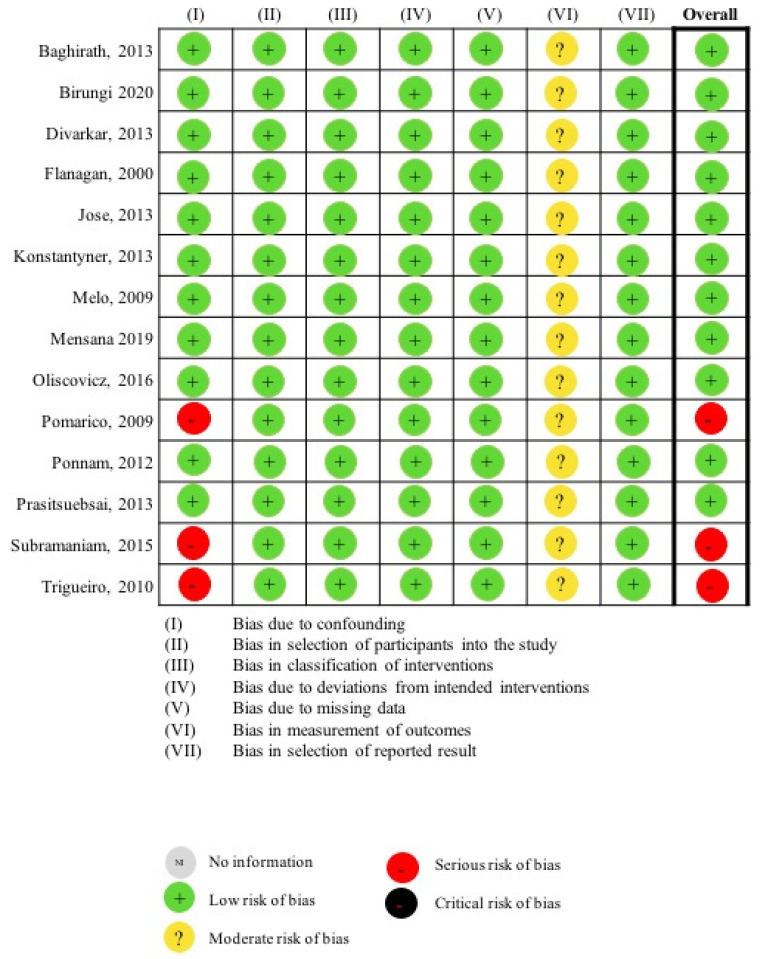
Risk of bias assessment with ROBINS-I [23,24,25,26,27,28,29,30,31,32,33,34,35,36].

**Figure 3 ijerph-19-11943-f003:**
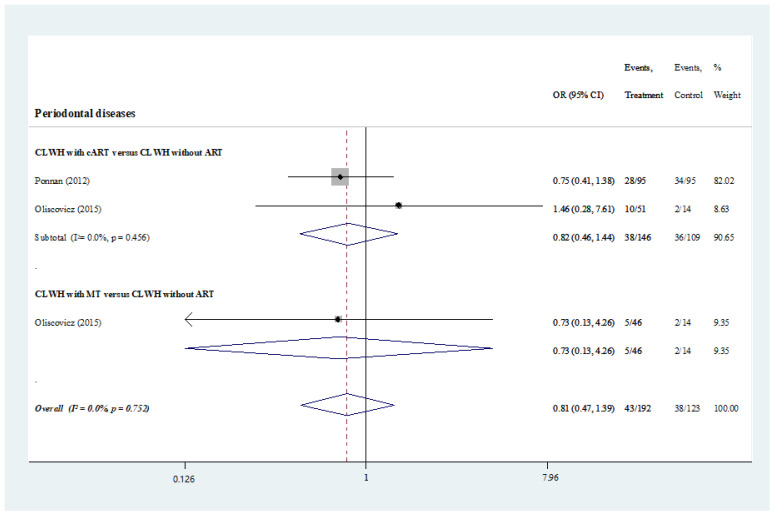
Prevalence of periodontal diseases. CLWH under ART versus those without medications [31,33].

**Figure 4 ijerph-19-11943-f004:**
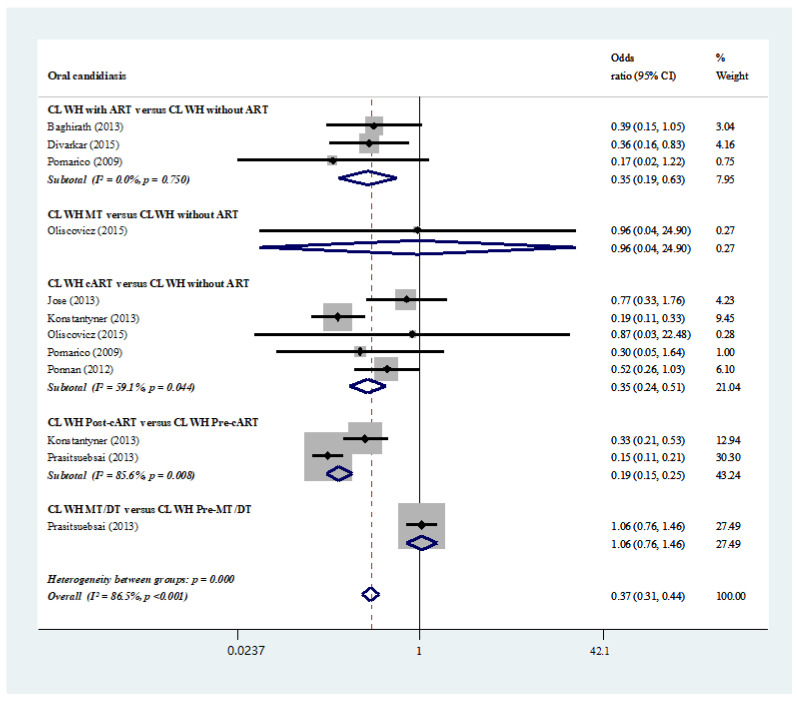
Prevalence of oral candidiasis: HIV-infected individuals under antiretroviral medications versus those without medications [23,25,27,28,31,32,33,34].

**Figure 5 ijerph-19-11943-f005:**
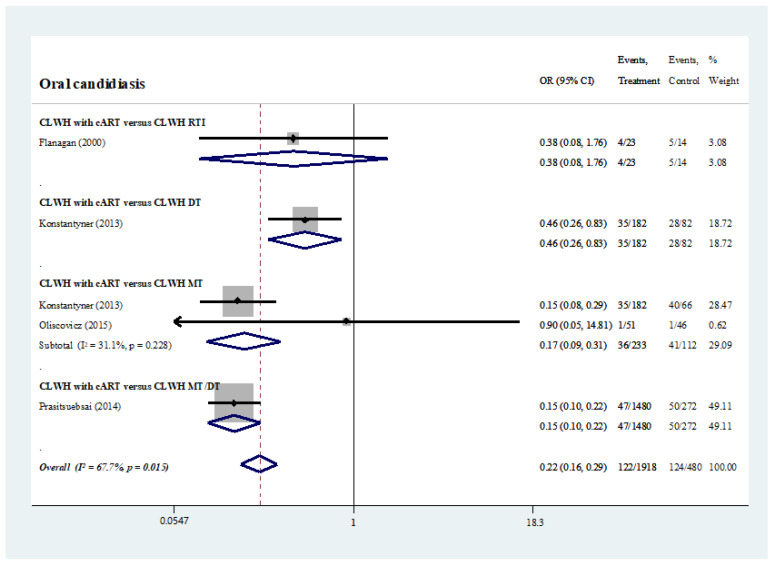
Prevalence of oral candidiasis: HIV-infected individuals under different antiretroviral medication [26,28,31,34].

**Table 1 ijerph-19-11943-t001:** Characteristics of included studies.

	Study (Year, Country ^a^) ^P/R^	Number of Subjects (% Males)		Age Range (Year)	Recruitment	Inclusion (I)/Exclusion (E) Criteria ^b^	Confounders Evaluated	Outcome Measures
1.	Baghirath (2013; IND) ^P^	CLWH ART	50 (46)	5–12	Nireekshana ART Centre, Hyderabad (NGO)	(I) Seropositive for antibody to HIV when tested by a particle agglutination test for antibodies to HIV and enzyme-linked immunosorbent assay (ELISA) (E) HIV-infected subjects with a history of local radiation therapy to the head and neck region.	✓ Duration of therapy	✓ Linear gingival erythema ✓ Hairy leukoplakia ✓ Angular cheilitis ✓ Oral ulcers ✓ Candidiasis ✓ Hyperpigmentation
		CLWH NMD	50 (50)	5–12				
2	Birungi (2020; UGA) ^P^	CLWH PI (Lopinavir/ Ritonanavir)	80 (48)	5–7	Ouagadougou University Teaching Hospital (urban site in Burkina Faso), East London Hospital Complex (urban site in South Africa); Mbale Regional Referral Hospital (semi-rural site in Uganda); and Lusaka University Teaching Hospital (urban site in Zambia).	(I) Singleton (I) Breastfed on day 7 by their mothers (I) Negative HIV-1 DNA PCR on day 7 (I) Received any prevention of mother-to-child treatment (E) Newborns who had clinical signs or biological abnormalities of grade 2 or higher on the US National Institutes of Health Division of AIDS adverse events grading tables (E) Serious congenital malformations or birthweight was 2.0 kg or lower.	✓ Marital status ✓ Socio-economic status ✓ Education level ✓ HIV clinical staging ✓ Detectable viral load ✓ Gender	✓ Caries ✓ Gingival bleeding
		CLWH NRTI (Lamivudine)	84 (53)	5–7				
3	Divakar (2015; SAU) ^P^	CLWH ART	62 (59.6)	5–15	District Hospital ART Centre	(I) Positive on particle agglutination test for antibodies to HIV and enzyme-inked immunosorbent assay (ELISA) (E) History of adverse habits such as tobacco and betel nut	✓ Duration of therapy	✓ Linear gingival erythema ✓ Hairy leukoplakia ✓ Angular cheilitis ✓ Oral ulcers ✓ Candidiasis ✓ Hyperpigmentation
		CLWH NMD	55 (53.7)	5–15				
4	Flanagan (2000, BRA) ^P^	CLWH cART	23 (NR)	6–18	University-Affiliated Paediatric Infectious Disease Clinic	(I) Active treatment at the paediatric HIV clinic (I) Ambulatory capability (I) Life expectancy of over three months	None	✓ Candidiasis
		CLWH RTI	14 (NR)	6–18				
5	Jose (2013, IND) ^P^	CLWH cART	47 (34.0)	2–12	Hospital and out-patient centres	(I) Definitive diagnosis for HIV infection confirmed either by enzyme-linked immunosorbent assay (ELISA)/Western blot/HIV tri-dot tests (I) A combination therapy or cART with nucleoside reverse transcriptase inhibitors such as combivir (zidovudine + lamivudine combination) and/or lamivudine + stavudine and non-nucleoside reverse transcriptase inhibitors such as nevirapine/efavirenz were the prescribed drug regimens. (E) Children requiring urgent medical attention	✓ CD4+ counts ✓ Duration of therapy	✓ Angular cheilitis ✓ Linear gingival erythema ✓ Recurrent oral ulcerations ✓ Oral candidiasis ✓ Oral hairy leukoplakia ✓ Chronic herpes simplex infection ✓ Hyperpigmentation
		CLWH NMD	53 (45.2)	2–12				
6	Konstantyner (2013; BRA) ^R^	CLWH Pre-cART	284 (NR)	0.3–17.8	Reference centre for the treatment of AIDS	(I) Laboratory confirmation of HIV infection contracted through vertical transmission (I) Less than 13 years of age at the time of admission to the health centre (E) Patients in follow-up for less than 90 days (E) Anti-inflammatory drugs and periodontal therapy in the past 6 months prior to the initial evaluation	✓ Antiretroviral regimen	✓ Oral candidiasis
		CLWH Post-cART	137(NR)	0.3–17.8				
7	Melo (2009; BRA) ^P^	CLWH MT	61 (80.3)	3–15	University Paediatric Immunodeficiency Outpatient Service		✓ CDC immunological category	✓ Oral candidiasis
		CLWH cART and PI	61 (80.3)	3–15				
8	Mensana (2019; IDN) ^P^	CLWH ART	24 (NR)	0–18	Outpatient clinic UPIPI, Soetomo General Hospital	(E) Parental refusal in providing consent (E) Children being not cooperative during examination (E) No CD4 values found within 6 months interval from the date of an oral examination on patients’ medical record	✓ Immune suppression ✓ Age	✓ Linear gingival erythema
		CLWH NMD	4 (NR)	0–18				
9	Oliscovicz (2016; BRA) ^R^	CLWH cART	51 (NR)	2–16	Paediatric AIDS Outpatient Clinic	(I) Definitively diagnosed with HIV infection according to criteria established by the Centers for Disease Control and Prevention	✓ None	✓ Gingivitis ✓ Parotid hypertrophy ✓ Linear gingival erythema
		CLWH ART	46 (NR)	2–16				
		CLWH NMD	14 (NR)	2–16				
10	Pomarico (2009; BRA) ^P^	CLWH cART	24 (50)	2–13	Paediatric AIDS Outpatients Clinic at the Federal University	(I) Definitive diagnosis for HIV infection confirmed by 2 positive ELISA tests and 1 positive Western blot (E) Presence of fixed or removable orthodontic appliances and systemic or local antifungal treatment within the last three months	✓ AIDS status	✓ Candida species ✓ Specific SIgA
		CLWH NMD	25 (42)	2–13				
		CLWH cART	16 (31)	2–13				
11	Ponnam (2012, IND) ^P^	CLWH cART	95(45.3)	5–15	Anti-Retroviral Therapy (ART) Center in Government General Hospital	(I) Came to the ART Center for the first time without previous history of anti-retroviral therapy	✓ Age ✓ Socio-economic status	✓ Candidiasis ✓ Caries ✓ Gingivitis/periodontitis ✓ Dental ✓ Ulcerative stomatitis ✓ Hyperpigmentation ✓ Mucocele
		CLWH NMD	95 (NR)	5–15				
12	Prasitsuebsai (2013, IDN, IND, KHM, MYS, THA,) ^P^	CLWH Pre-cART	2129 (NR)	0–18	14 participating clinics in Cambodia (n = 3), India (n = 1), Indonesia (n = 1), Malaysia (n = 4) and Thailand (n = 5).	(I) Conclusively diagnosed with HIV, using age-appropriate testing or through a presumptive clinical diagnosis of HIV infection defined as meeting WHO criteria for initiating antiretroviral therapy (ART). (I) Retrospective data provided were available.	✓ CD4+ counts	✓ Oral candidiasis ✓ Acute necrotising ulcerative ✓ Gingivitis/periodontitis ✓ Kaposi’s sarcoma
		CLWH ART	1480 (50.5)	0–18				
		CLWH MT, DT	272 (51.8)	0–18				
13	Subramaniam (2015; IND) ^P^	CLWH Pre-ART	25 (80)	6–8	HIV centres	(I) HIV-infected children aged 6–8 years (I) Children prior to the onset of anti-retroviral therapy and Group 2 consisting of children undergoing anti-retroviral therapy for more than 3 years	None	✓ Specific SIgA
		CLWH Post-ART	25 (58.5)	6–8	HIV centres			
14	Trigueiro (2010; BRA) ^R^	CLWH NRTI	6 (NR)	3–14	University Centre for Special Care Patients	(I) HIV-positive patients	None	✓ Dental mineralisation chronology
		CLWH NRTI+PI	23 (NR)	3–14				
		CLWH NRTI+NNRTI	11 (NR)	3–14				
		CLWH NRTI+NNRTI+PI	6 (NR)	3–14				
		CLWH NMD	7 (NR)	3–14				

^a^ ISO alpha-3 codes of countries; ^b^ extracted and quoted from the included articles; ^P^ prospective study; ^R^ retrospective study. Note. ART = antiretroviral therapy; Cart = combined antiretroviral therapy; CLWH = children living with HIV; MT = monotherapy; NR = not reported; NRTI = nucleoside reverse transcriptase inhibitor; NNRTI = non-nucleoside reverse transcriptase inhibitors; NMD = no medication; PI = protease inhibitors; RTI = reverse transcriptase inhibitors.

## Data Availability

The data presented in this study are available in the Appendix A.

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
