# Peer review of "Impact of Antiretroviral Therapy on Oral Health among Children Living with HIV: A Systematic Review and Meta-Analysis"

_ijerph, 2022, doi:10.3390/ijerph191911943_

Round 1

Reviewer 1 Report

Dear Authors,

I have read you manuscript titled "Impact of antiretroviral therapy on oral health among children living with HIV: a systematic review and meta-analysis" with interest.

The work describes a meta-analysis of oral health of children with HIV undergoing ART.

Overall I feel the paper is well presented and really there are no major flaws that I can see, within the limitations of the literature.

There are a few minor points:

1. The English is of a good standard although I have noticed a few mistakes. For example, in line 21 of the abstract it says "Children undergoing CLWH" which I think perhaps should be "Children undergoing ART"? I'm not sure it makes sense otherwise. I would advise careful proof-reading for any such typos.

2. You mention using funnel plots to check for bias in the methods, though these are not presented even in supplementary data. Can these be included? 

3. Much of the data would be better presented graphically. If possible within the confines of the journal, can supplementary figures 1, 2 and 4 be put in the main body of the text?

4. The introduction may benefit from a section describing the typical HIV-associated oral conditions for readers less familiar with these.

Reviewer 2 Report

The manuscript "Impact of antiretroviral therapy on oral health among children living with HIV: a systematic review and meta-analysis" presents a systematic review summarizes the current evidence and compares the oral health status of CLWH (children living with HIV) who were treatment-naïve with those undergoing different ART (antiretroviral therapy) medications.

In the manuscript, the question is original and well-defined and the results provide an advancement of current knowledge and insights into future research. The topic, very interesting, can attract a large audience of readers.

The study is adequately designed, technically valid and drawn up according to the highest technical standards. The data and analyzes are presented appropriately, as is the statistical analysis. The care with which useful and scientifically significant data was filtered and selected was remarkable. The results are interpreted appropriately. All conclusions are justified and supported by the results.

The article is written correctly, except for some typos. I recommend a review of the general structure (paragraphs, numbering, tables) and the updating of the bibliography with sources that I consider useful for addressing the topic in the introductory phase that I am attaching to the authors.

I believe there is a general benefit to publishing this work.

The systematic review and meta-analysis is carefully written and scientifically valid. The data is analyzed appropriately. The results consistent with the evidence found. Useful conclusions for clinicians and for future research ideas.

Therefore, I recommend that you:

- check grammatical and typing errors;

- revise the formatting (paragraph numbering, tables)

- I believe it may be appropriate to mention the following studies in the introductory part to the topic under consideration: 1) Healthcare (Basel). 2022 Feb 6; 10 (2): 308. doi: 10.3390 / healthcare10020308. 2) Healthcare (Basel). 2022 Jun 26; 10 (7): 1195. doi: 10.3390 / healthcare10071195.

Best Regards
